# Macronutrients and Micronutrients in Parenteral Nutrition for Preterm Newborns: A Narrative Review

**DOI:** 10.3390/nu14071530

**Published:** 2022-04-06

**Authors:** Valentina Rizzo, Manuela Capozza, Raffaella Panza, Nicola Laforgia, Maria Elisabetta Baldassarre

**Affiliations:** 1Intensive Care Unit, Department of Biomedical Science and Human Oncology (DIMO), Section of Neonatology and Neonatal, 70124 Bari, Italy; manuelacapozza26@gmail.com (M.C.); mariaelisabetta.baldassarre@uniba.it (M.E.B.); 2Neonatology and Neonatal Intensive Care Unit, A. Perrino Hospital, 72100 Brindisi, Italy; 3Section of Neonatology and Neonatal Intensive Care Unit, Interdisciplinary Department of Medicine (DIM), University of Bari Aldo Moro, 70124 Bari, Italy; nicola.laforgia@uniba.it

**Keywords:** parenteral nutrition (MeSH), macronutrients (MeSH), micronutrients (MeSH), newborns, infants (MeSH), prematurity (MeSH)

## Abstract

Preterm neonates display a high risk of postnatal malnutrition, especially at very low gestational ages, because nutritional stores are less in younger preterm infants. For this reason nutrition and growth in early life play a pivotal role in the establishment of the long-term health of premature infants. Nutritional care for preterm neonates remains a challenge in clinical practice. According to the recent and latest recommendations from ESPGHAN, at birth, water intake of 70–80 mL/kg/day is suggested, progressively increasing to 150 mL/kg/day by the end of the first week of life, along with a calorie intake of 120 kcal/kg/day and a minimum protein intake of 2.5–3 g/kg/day. Regarding glucose intake, an infusion rate of 3–5 mg/kg/min is recommended, but VLBW and ELBW preterm neonates may require up to 12 mg/kg/min. In preterm infants, lipid emulsions can be started immediately after birth at a dosage of 0.5–1 g/kg/day. However, some authors have recently shown that it is not always possible to achieve optimal and recommended nutrition, due to the complexity of the daily management of premature infants, especially if extremely preterm. It would be desirable if multicenter randomized controlled trials were designed to explore the effect of early nutrition and growth on long-term health.

## 1. Introduction

The incidence of preterm birth is on the rise: 11% of births worldwide occur before the 37th week of gestation. Survival of preterm infants continues to increase both in developed and developing countries, and in part reflects improvements in nutritional care. However, the preterm population presents a high risk of postnatal malnutrition, and this risk is higher for infants of low gestational age (GA) because nutritional reserves are lower in younger preterm infants. For this reason several studies continue to demonstrate the importance of nutrition and growth in early life.

Preterm birth interrupts the physiological growth path of the fetus that occurs during the third trimester of pregnancy. Preterm infants are exposed to an environment which is different from the maternal uterus, hence requiring more energy to maintain thermal and metabolic homeostasis. Extremely preterm infants (less than 28 weeks’ gestation) also feature little or no subcutaneous fat and low stores of glycogen and key nutrients such as iron, zinc, calcium, and vitamins. Therefore, the nutritional care for preterm newborns remains a challenge in clinical practice since an inadequate nutrition in the first days of life could determine not only growth failure [1], but also increased vulnerability to infectious diseases arising from suboptimal immune defenses, along with severe respiratory distress and compromised functionality of other organs such as intestine, skeletal muscle, and brain.

Some authors have recently shown that it is not always possible to achieve optimal and recommended nutrition, due to the complexity of the daily management of preterm infants, especially if extremely preterm. Infusion of drugs, maintenance of vascular access, and volume boluses to support blood pressure commonly result in administration of relatively large volumes of non-nutritional fluids [2].

## 2. Methods

This narrative review was conducted by searching scientific databases for articles reporting on parenteral nutrition in preterm neonates and on correct macronutrient and micronutrient intake.

To fully investigate the importance of parenteral nutrition in preterm neonates, an exhaustive search for eligible studies was performed in PubMed, Embase, Medline, Cochrane library and Web of Science databases.

The following subject MeSH headings were used: “Parenteral nutrition” (MeSH), “Macronutrients” (MeSH), “Micronutrients” (MeSH), “Energy intake” (MeSH), “Infants, Newborn” (MeSH), “Premature Birth” (MeSH), and “Prematurity” (MeSH).

To be as comprehensive as possible, proper Boolean operators such as “AND” and “OR” were also included. Additional studies were sought using references in articles retrieved from searches.

We included all relevant articles (RCTs, observational studies, guidelines and reviews) written in English, involving only human subjects, and published between January 2011 and November 2021.

## 3. Results and Discussion

### 3.1. Energy Intake

Preterm neonates are exposed to a high risk of faltering growth, mainly due to undernutrition. In order to limit growth retardation, current guidelines for preterm neonates recommend the administration of high doses of protein and energy intakes through parenteral nutrition (PN), starting soon after birth [3]. Early postnatal growth failure describes the severe nutritional deficit that develops in preterm infants in the first few weeks of life. The deficit refers to the gap between energy, protein and other nutrient requirements needed to mimic fetal growth rates and the actual supply delivered to preterm infants. Recent recommendations suggest a calorie intake of 120 kcal/kg/day and a minimum protein intake of 2.5–3 g/kg/day [4].

Kashyap et al. showed that greater protein intakes would increase weight, length, and head circumference, whereas additional lipids only increased weight and subcutaneous fat. Such diets lead to fatter, shorter, less muscular infants, and perhaps to long-term neurological deficits. This issue is crucial for the smallest, most preterm infants who are fed far more energy and much less proteins than needed to meet appropriate growth rate and body composition. Boscarino et al. [5] in 2021 demonstrated that the route of energy administration has different impacts on cerebral growth. Specifically, administration of high energy intakes through the enteral route has positive effects on the growth of transverse cerebellum diameter, caudate head and cerebellar vermis height at neonatal age.

However, Bonsante et al. demonstrated in an observational study that high PN energy intake is associated with metabolic acidosis [6]. Moreover, high energy intake in early life may favor hyperglycemia which is in turn associated with mortality and cerebral impairment of survived preterm newborns [5].

### 3.2. Liquids

Despite the fact that hundreds of thousands of preterm infants receive parenteral fluids each year, study of optimal fluid and electrolyte management in this population is limited. Appropriate fluid and electrolyte management is critical for optimal care of low birth weight or sick infants, as fluid overload and electrolyte abnormalities pose significant morbidity threats. Water and electrolyte balance is influenced by different factors, such as insensible cutaneous and respiratory water loss, contraction of the extracellular compartment in the first days of life, renal immaturity, and needs related to growth. Therefore, water intake must compensate for ‘renal excretion and water loss through the skin and respiratory and digestive tracts, in order to ensure adequate growth [7].

Nonetheless, the right and appropriate volume of fluids to administer to premature newborns is not yet fully defined. It is important to define the right fluid intake because an adequate intake in preterm newborns is essential to prevent deficits and promote growth, improving the potential outcome for the most fragile subjects.

Restrictive fluid administration significantly decreases the risk of patent ductus arteriosus (PDA) and necrotizing enterocolitis (NEC) [8], albeit increasing postnatal weight loss and risk of dehydration, malnutrition, hypotension, peripheral hypoperfusion, renal damage, hypoglycemia, hyperosmolality, and hyperbilirubinemia [9].

In the first days of life water intake should be individualized, considering changes in the neonate’s weight, serum sodium and urine output. Fluid needs in preterm neonates are postnatal and gestational age dependent. At birth, water intake is 70–80 mL/kg/day and progressively increases, reaching 150 mL/kg/day at the end of the first week of life [7]. Fluid requirements in the first day of life are usually higher for the lower gestational age, reaching up to 100 mL/kg/day in ELBW infants [10]. Water intake can reach up to 200 mL/kg/day, depending on variables such as neonate’s characteristics, thermal environment, sodium intakes, type of solution, clinical and biological parameters.

Further research is needed to clarify the optimal fluid regimen for preterm infants to ensure adequate hydration and nutrition without contributing to serious complications.

#### Electrolytes

Electrolytes intake is also very important for the adequate growth of preterm infants, in particular sodium balance, since wide variations in serum sodium are associated with a worse prognosis at 2 years of life [11]. For any infant, regardless of gestational age, renal adaptation to the extrauterine environment is necessary, which includes the ability to concentrate urine and retain body water. Infants born prematurely have higher total body water (TBW) and extracellular water (ECW) per kilogram than do term infants. The degree of contraction of the ECW compartment is inversely proportional to gestational age and greater in small for gestational age than appropriate for gestational age infants. Preterm infants may exhibit a 10–15% weight loss during the first week of life related to loss of ECW, whereas term infants generally exhibit a 5–7% weight loss [12].

At birth there is a physiological loss of sodium because of the normal contraction of extracellular volume in the first days of life. Subsequently, after the initial relative oliguria, a diuretic phase occurs which can lead to sodium loss due to increased urine output. Hence, after a first restriction of Na intake (less than 2 mEq/kg/day at birth), a rapid increase in sodium intake (4 mEq/Kg/day, or adapted to sodium losses) must be considered [13]. Current recommendations for preterm infants from the American Academy of Pediatrics (AAP) and ESPGHAN include enteral or parenteral intake of sodium of 3–5 mEq/kg/day during the stable, growth phase of postnatal care. Studies in preterm infants suggest that sodium supplementation may optimize weight gain. Vanpee et al. randomized ten infants with a gestational age of 29–34 weeks to receive sodium supplementation of 4 mEq/kg/d from 4–14 days of life. At 2 weeks of age, supplemented infants weighed about 6% above birthweight while non-supplemented infants were a little less than 2% below birthweight. A few years later similar results were demonstrated by Isemann et al. They randomized infants < 32 weeks gestation to receive 4 mEq/kg/d of sodium or placebo from 7–35 days of life, with an average daily sodium intake of 6.3 mEq/kg in the supplemented group and 2.9 mEq/kg in the placebo group. At six weeks of postnatal age, 79% of supplemented infants maintained their birthweight percentile compared with only 13% in the placebo group. No significant differences in length or head circumference were identified. These studies suggest that avoiding sodium deficiency in preterm infants may result in improved weight gain.

Plasma potassium concentration is higher in preterm than in term neonates, and it typically falls in the first few days of life [14]. For this reason potassium supplementation should not begin until adequate urine output is established. The recommended intake is 1–2 mEq/kg/day [13]. Potassium is the most abundant cation of intracellular fluids and plays an important role in numerous intracellular functions [15]. Non-oliguric hyperkalemia is frequently seen in very low birth weight infants in the first 48 h after birth, despite minimal potassium intake. Elevated plasma potassium levels are likely to result from a shift from the intracellular to the extracellular space, in part due to decreased Na-K-ATPase activity, and limited renal potassium excretion associated with a low glomerular filtration rate [16]. Kalemia reflects not only potassium intake, but also potassium flow to the extracellular compartment, and potassium secretion in urine and intestinal lumen. It is important to guarantee a higher and earlier aminoacidic and energy intake to avoid non-oliguric hyper-kalemia and improve potassium balance [17].

Calcium and phosphate (Ca-P) should be added from the first day; 60–80 mg/kg of elemental Ca, 45–60 mg/kg of phosphate per day should be given [18]. During the first days of life a molar calcium-to-phosphorus ratio at 1 or below prevents early hypophosphatemia.

However, high aminoacidic and energy intake can also cause hypophosphatemia, hypercalcemia, and hypercalciuria [19]. Chlorine is generally taken passively when potassium and phosphorus are prescribed in the form of chlorides, with frequent excessive intake [20,21]. The use of other sodium, potassium, or phosphorus salts can reduce the risk of metabolic acidosis. In general, total chloride intake should be lower than the addition of sodium and potassium intakes in order to avoid hyperchloremic metabolic acidosis [7].

### 3.3. Proteins

Proteins are the major structural and functional component of all cells in the body and consist of chains of amino acids joined together by peptide bonds.

Providing adequate protein supply to preterm infants during the early postnatal period is critical to ensure proper growth and avoid early postnatal malnutrition, that may affect long-term neurodevelopmental outcome.

A minimum intake of 1.5 g/kg/day intravenous protein is required to prevent a negative nitrogen balance in preterm babies and this forms the basis of current recommendations for intravenous amino acid intakes for VLBW babies (starting dose of 1.5 g/kg/day with progressive daily increases up to 2.5–3.5 g/kg/day in the first week of life) [22] (Table 1).

**Table 1 nutrients-14-01530-t001:** Proteins.

First Author	Type of Study	Sample Size (*n*)	Intervention	Outcomes
van Goudoever JB [22] 2018	ESPGHAN/ESPEN/ESPR/CSPEN guidelines		starting dose of 1.5 g/kg/day with progressive daily increases up to 2.5–3.5 g/kg/day in the first week of life	
Roelants JA [23] 2018	RCT	32	initiation of amino acids (2.4 g/kg/day)	improves the net amino acid balance
Morgan C [24] 2014	RCT		higher protein intakes	Improves brain growth and neurodevelopmental outcomes

RCT (Randomized Controlled Trial).

Brain growth and development depend on a high rate of protein synthesis because protein is essentially the structural scaffolding of the brain, as shown by the positive correlations between higher protein intakes, greater head circumference growth and improved neurodevelopmental outcomes [24]. Morgan et al. compared parenteral nutrition supplying 2.8 g/kg/day amino acids (and 2.8 g/kg/day lipid) vs. 3.8 g/kg/day amino acids (and 3.8 g/kg/day lipid) from 6 h of life in premature infants under 1200 g and 29 weeks of gestational age [25].

The growth of head circumference, an index of brain growth correlated with neurodevelopment, between day 1 and day 28 was higher (with a gain of 0.37 Z-score) with high amino acid intake.

In a randomized prospective study, Roelants et al. demonstrated that starting amino acid intake in the early hours of life does not result in hyperammonemia or acidosis, and improves the net amino acid balance as measured by perfusion of 13C-leucine. Compared to glucose alone, the initiation of amino acids (2.4 g/kg/day) in the first 2 h of life in 32 premature babies with an average weight of 923 g improved not only protein synthesis and nitrogen balance on the 2nd day of life, but also the synthesis of glutathione [23].

### 3.4. Carbohydrates

Glucose is the primary source of non-protein energy in PN and is essential for the developing brain after birth because it is easily metabolized. Preterm neonates have low stores of glucose in the form of glycogen, hence an adequate source of exogenous glucose is mandatory to ensure satisfactory growth and neurological development [10].

Glucose requirements in the neonatal period vary according to gestational age and the individual needs of the neonate. For most infants, a glucose infusion rate of 3–5 mg/kg/min is sufficient, but VLBW and ELBW preterm neonates may require up to 12 mg/kg/min to maintain enough energy for their metabolism. Correct glucose intake must be provided to guarantee adequate growth rates, in particular to support growth and protein synthesis from amino acids [26].

In preterm neonates glucose production rates is 2–3 mg/kg/min, and glucose utilization rates can be as high as 7–9 mg/kg/min, for the brain but also for the heart. Term infants also produce 2–3 mg/kg/min of glucose, but their glucose utilization rates are lower (3–5 mg/kg/min) [26] (Table 2).

**Table 2 nutrients-14-01530-t002:** Carbohydrates.

First Author	Type of Study	Sample Size (*n*)	Intervention	Outcomes
Hay WWJ [26] 2017	Narrative review		preterm neonates may require up to 12 mg/kg/min	to maintain enough energy for their metabolism.
Ribed Sánchez A [27] 2013	Observational study	68	aggressive nutrition (the use of high nutrient dosages starting in the first hours of life, in particular the administration of protein and energy at higher concentrations) than conventional parenteral nutrition	To decrease the frequency and severity of neonatal hyperglycemia by stimulating endogenous insulin secretion and promotes growth by stimulating Insulin-Growth Factors
Angelika D [28] 2021	Observational study	97	the GIR (glucose infusion rate) usage of <7 g/kg/day in PN in the first week of life for preterm neonates	To increase the risk of hypoglycemia while reducing the risk of sepsis
Tottman AC [29] 2018	Observational study	457	lower glucose administration (7 versus 8.4 mg/kg/min)	To reduce the risk of sepsis

In preterm infants it is important to avoid not only hypoglycemia but also hyperglycemia which is a cause of hypoxia, acidosis, and various forms of toxicity inducing cellular and systemic inflammation [30].

Glucose should be infused to meet needs, adjusting the glucose infusion rate (GIR) to maintain normal glucose concentrations (50–100 mg/dL). Maximum intravenous glucose infusion rates should not exceed 6–10 mg/kg/min (equivalent to 27–42 kcal/kg/day).

Recently the concept of “aggressive nutrition” has emerged, with early administration of lipids and a significant supply of protein along with high glucose from the first hours of life [31]. In particular, by aggressive nutrition we mean the use of high nutrient dosages starting in the first hours of life, in particular the administration of protein and energy at higher concentrations than in the previous conventional PN [32]. Early administration of proteins decreases the frequency and severity of neonatal hyperglycemia by stimulating endogenous insulin secretion and promotes growth by stimulating Insulin-Growth Factors [27]. This PN practice must be administered directly after birth to promote optimal plasma glucose levels and to ensure a positive energy balance. In early aggressive PN, GIR is enhanced up to 12.5 mg/kg/min of glucose to attain optimal nutritional support which aims to improve growth and development outcomes [33].

Evidence has demonstrated that the introduction of amino acids within 4 h after birth diminishes insulin-treated hyperglycemia incidence in preterm infants.

On the other hand, some studies have shown that decreasing carbohydrate intake to 7 mg/kg/min reduced the risk of neonatal hyperglycemia in preterm infants [29].

More than half of preterm neonates require a continuous glucose infusion immediately after birth to maintain blood glucose levels [34].

The GIR for preterm infants in PN should be maintained at 6–8 mg/kg/min to ascertain adequate glucose requirements [35]. Some literature provides recommendations for glucose administration of PN in preterm infants with various GIRs, in expressing carbohydrate intakes such as GIR of 5.5–8 mg/kg/min [36], 4–10 mg/kg/min [33], or 5.5–12.5 mg/kg/min [37]. The most likely explanation for this wide variation in GIRs is that the value of the maximum glucose oxidase capacity is not completely known in neonates. The rate of glucose administration for parenteral nutrition should exceed the maximum glucose oxidase capacity, which is possibly as high as 12.5 mg/kg/min [28].

Early aggressive parenteral nutrition potentially increases the risk of hyperglycemia [30], especially with the use of GIR > 10 mg/kg/min [38].

In a recent study Angelika et al. showed that the GIR usage of <7 g/kg/day in PN in the first week of life for preterm neonates was an independent variable significantly increasing the risk of hypoglycemia while reducing the risk of sepsis [28]. These findings are consistent with a study by Tottman et al. which reported that a lower glucose administration reduced the risk of sepsis (7 versus 8.4 mg/kg/min) [29].

However, further research is needed to better define glucose administration in PN in ELBW preterm infants. How much glucose is optimal to provide a better outcome in preterm infants receiving parenteral nutrition is unclear and the results of related studies are conflicting.

### 3.5. Lipids

Lipids are a pivotal source of energy and calories for growth, but also provide essential fatty acids which are required for brain development. Hence, intravenous lipid emulsion in PN is crucial to prevent essential fatty acid deficiency, which is frequent in preterm babies due to the shortage of stores. Intravenous lipid emulsions are linked to critical outcomes such as neurodevelopment, anthropometric measures (weight gain, linear growth and head circumference) and adverse effects of lipids, including sepsis and PN related liver disease (abnormal liver function, cholestasis and raised conjugated hyperbilirubinemia).

Pooled data revealed that no significant differences were found comparing high (3–3.5 g/kg/day) versus low (1 g/kg/day) target dose of lipids in terms of sepsis [39,40], cholestasis [39,40], mortality [40,41] and length of stay [39,40,41].

In comparison with a lower dosage of lipids, higher target dosage of lipids was associated with increased mean weight gain in the first 28 days [39], and decreased rates of necrotizing enterocolitis and retinopathy or prematurity [40]. Therefore, it is recommended to reach a target dosage of lipids of 3–4 g/kg/day at maximum, which was proven to be safe and effective [42,43].

Studies across the years have assessed two starting dosage thresholds: 0.5–1 g/kg/day was chosen as the lower starting dosage whereas 2 g/kg/day was selected as the upper starting dosage threshold (Table 3).

**Table 3 nutrients-14-01530-t003:** Lipids.

First Author	Type of Study	Sample Size (*n*)	Intervention	Outcomes
Calkins KL [39] 2017	RCT	41	high (3–3.5 g/kg/day) versus low (1 g/kg/day) target dose of lipids	no significant differences in terms of sepsis, cholestasis, mortality. Increased mean weight gain in the first 28 days
Levit OL [40] 2016	RCT	127	high (3–3.5 g/kg/day) versus low (1 g/kg/day) target dose of lipids	no significant differences were found in terms of sepsis, cholestasis, mortality and length of stay. Decreased rates of necrotizing enterocolitis and retinopathy or prematurity
Lapillonne A [43] 2018	LG ESPGHAN		target dosage of lipids of 3–4 g/kg/day at maximum	Safe and effective
NICE Guideline [42] 2020			target dosage of lipids of 3–4 g/kg/day at maximum	Safe and effective
Vlaardingerbroek [44] 2013	RCT	144	comparing preterm babies started early (i.e., soon after birth) versus late (i.e., on day 2 of life) on lipid emulsions	no significant differences in anthropometric measures at discharge, late onset sepsis, necrotizing enterocolitis, retinopathy of prematurity and mortality rates

Slowly increasing lipids from a low starting dose (e.g., 0.5 g/kg/day) to a target dose (e.g., 3 g/kg/day), using an infusion rate of 0.5 g/kg/day, may be associated with a reduced number of babies with retinopathy of prematurity and hypertriglyceridemia compared with those who start at a higher dosage (2 g/kg/day) and reach the same target dosage in a shorter time. Conversely, a benefit for the number of babies who are equal to or greater than the 10th percentile for weight was found in the higher starting dosage and shorter time to target dosage [45].

Kao et al. showed reduced rates of sepsis in preterm babies with continuous lipid delivery (as per intermittent infusion, but over 24 h per day) compared to intermittent delivery (8 h/day lipid infusion (Intralipid 10%) at a starting dose of 0.5 g/kg/day, increasing incrementally by 0.5 g/kg/day to either 3 g/kg/day or until fat contributed to 40% of daily calories) [46].

Therefore, it is recommended to start lipids from a low starting dose (e.g., 0.5 g/kg/day) and slowly increase to a target dose in the maintenance range (e.g., 3 g/kg/day), according to the baby PN tolerance. It is advisable to increment by 0.5 g/kg/day and favor continuous infusions [42,43].

As for timing of delivering lipids, current practice is to start infusion soon after birth. When comparing preterm babies started early (i.e., soon after birth) versus late (i.e., on day 2 of life) on lipid emulsions, Vlaardingerbroek et al. showed no significant differences in anthropometric measures at discharge, nor late onset sepsis, necrotizing enterocolitis, retinopathy of prematurity and mortality rates [44].

However, delaying lipids causes fatty acid deficiency within the first two days of life in the vulnerable preterm population. Although some authors have shown an increased risk of cholestasis when lipids are started soon after birth [47], there was also some evidence of slightly reduced retinopathy of prematurity [44] and improved neurodevelopmental outcomes [23].

Therefore, it is currently recommended to start lipid emulsions soon after birth [42,43].

In case of babies starting PN after the first 4 days of life, lipids and other macronutrients should be started, based on the recommended maintenance range and adjusted according to enteral feed intake and tolerance [42].

Lipid 20% emulsions are better tolerated than 10% due to the lower phospholipid concentration, with the same content of triglycerides. The excess of phospholipids leads to the formation of liposomes which, by accumulating free cholesterol, albumin and apoproteins, form the Lipoprotein X capable of interfering with the action of lipoprotein lipase by competition with triglycerides and LDL [48,49,50,51].

Heparin frees endothelial lipoprotein lipase, significantly increasing its circulating levels; however, it also causes an increase in triglycerides, cholesterol and fatty acids, most of which are not used. Currently, there are no convincing elements to suggest the use of heparin with the aim of improving the clearance of lipids [52].

Carnitine facilitates the transport of long-chain fatty acids across the mitochondrial membrane, improving the possibility of oxidation. Premature babies feature carnitine low deposits and limited synthesis capacity; however, in vitro studies have shown that β-oxidation of fatty acids is impaired only when tissue carnitine levels are below 10% of normal values [53].

Moreover, a systematic review of six randomized controlled trials in which carnitine was added to parenteral nutrition in doses ranging from 8 to 24 mg/kg/day, for a period ranging from 6 days to 40 weeks of post-conceptional age, showed no difference compared to the control group in terms of weight growth and lipid tolerance [54].

It is possible that greater intakes of carnitine are required, taking into account that a dosage of 48 mg/ kg/day has been associated with an increased metabolism with reduced storage of fats and proteins [55].

The administration of carnitine could be useful in particular conditions such as in subjects in prolonged total PN and in SGA infants, generally excluded from the populations studied.

There are numerous reports of the potential damage of lipid emulsions in the lungs of the preterm infant. Most of these observations, however, are based on weak study methodology, or far from clinical routine [47,56,57,58,59,60]. A more cautious use of intravenous lipids in conditions of pulmonary hypertension has been suggested, justified by an increase in pulmonary vascular resistance, mediated by thromboxane. Even in this case, however, the report is based on biochemical and instrumental data which are not associated with substantial changes in clinical and blood gas analytical parameters [61].

The use of lipids during jaundice has been much debated: the fatty acids released during hydrolysis could displace bilirubin from the albumin carrier, increasing the risk of nuclear jaundice. In vivo studies have shown that a molar ratio of less than 6 between free fatty acids (FFA) and albumin is not associated with bilirubin displacement [62]. In clinical practice, with the generally used intake of lipids, it is very unlikely that a molar ratio of FFA/Albumin greater than 3 will be reached [63,64,65]. There is also no literature data that has attributed an increased incidence of nuclear jaundice to the use of intravenous lipids in the premature baby.

Lipid emulsions, containing a high proportion of polyunsaturated fatty acids, are highly susceptible to peroxidation, the products of which can in theory trigger tissue damage from free radicals [66,67,68], since lipo-peroxides can significantly increase with exposure to ambient light (three times) and phototherapy (13 times) [69,70]. It is always advisable to screen the bag and the infusion line [71].

### 3.6. Trace Elements

Infants born prematurely are at increased risk of trace-mineral deficiencies because of low stores at birth, very rapid postnatal growth and variable intake. Although some trace elements (TE) may not be required for short-term PN therapy, all patients requiring long-term parenteral nutrition should be provided with TEs [72].

### 3.7. Zinc

Zinc (Zn) is an essential nutrient, involved in the metabolism of energy, proteins, carbohydrates, lipids and nucleic acids and is crucial for tissue accretion. Infants on long term PN may commonly have zinc deficiency and may have growth retardation, infections and a typical skin rash [73]. Children with increased enteral fluid losses are at especially high risk. Urinary Zn excretion and enteral Zn losses occur in parenterally fed infants. Some amino-acids such as histidine, threonine, and lysine bind Zn, increasing its renal ultra-filterability. Increased urinary losses of Zn and decreased plasma concentrations occur following thermal injury in children. Current recommendations are to supply 400–500 µg/kg/day in preterm infants, 250 µg/kg/day in infants from term to 3 months, 100 µg/kg/day for infants from 3 to 12 months and 50 µg/kg/day in children >12 months of age (up to a maximum of 5 mg/day for routine supplementation) [74].

### 3.8. Copper

Copper (Cu) is an essential nutrient, and is a functional component of several enzymes, including cytochrome oxidase, superoxide dismutase, monoamine oxidase and lysyl oxidase. Cu deficiency, which is associated with pancytopenia and osteoporosis, has occasionally been reported in children on long term PN [73]. Parenteral Cu requirements are estimated to be 40 µg/kg/day Cu for preterm infants and 20 µg/kg/day for term infants and children [75]. Patients with high gastro-intestinal losses, such as jejunostomies, external biliary drainage, exudative burns or continuous renal replacement therapy also have high Cu losses [76]. In severe Cu deficiency, serum Cu and ceruleo-plasmin levels are low and reflect the Cu status of the body. Preterm infants have high copper requirements but are also at risk of biliary stasis [77]. The European Society of Pediatric Gastroenterology, Hepatology and Nutrition (ESPGHAN) and the European Society of Parenteral and Enteral Nutrition (ESPEN) 2018 guidelines recommend a doubling of the provision of Cu in PN to preterm infants (from 20 to 40 µg/kg/day) [72].

### 3.9. Selenium

Pediatric recommendations for selenium provision in PN, except for preterm infants, are consistent across publications at 2–3 µg/kg/day with a maximum of 60 to 100 µg/day [72]. Critically ill patients or those with severe burns may have higher requirements due to increased oxidative stress and losses through drains, dialysis, or wounds. Doses of 3 µg/kg/day may prevent a decline in cord levels and doses of up to 5–7 µg/kg/day may be required to achieve concentrations close to those found in healthy breast fed infants. Parenteral supplementation of selenium to preterm infants significantly reduced the risk of one or more episodes of sepsis but was not associated with improved survival, reduction in chronic lung disease, or retinopathy of prematurity.

Although there are no reports of selenium toxicity in children [72], caution is required in case of renal failure as kidneys are the major route of excretion of selenium [78].

### 3.10. Iodine

European and Australian, unlike American, trace element commercial solutions usually contain iodine. A parenteral dose of 1 µg/kg/day of iodine is suggested for children by most nutrition societies. There are pediatric reports of iodine deficiency and hypothyroidism described with iodine-free PN use [79], highlighting the importance of iodine substitution given its impact on neurodevelopment.

### 3.11. Fluoride

Information on fluoride provision in PN in children is scarce, perhaps because fluoride deficiency has not been described in children on PN. Fluoride is not considered as an essential element, although it may contribute to bone strength and prevention of dental caries. ESPGHAN/ESPEN guidelines do not recommend fluoride supplementation of PN for children [72].

### 3.12. Vitamins

Vitamins are essential for growth and development especially in premature infants on parenteral nutrition. The needs and biological role of individual vitamins in preterm infants have been the subject of various critical reviews and some studies, which generally agree on the following points:-the pharmacological addition of vitamins is not required, e.g., high doses of vitamins A and E are not proven as effective in minimizing oxidative phenomena or incidence and severity of chronic lung diseases, retinopathies and other diseases.-vitamins should be used in adequate doses to meet normal needs, avoiding both deficiencies and potentially harmful excesses.

A physiological vitamin intake must be guaranteed early to all preterm VLBWs, possibly already within the first week of life. It is advisable to use specifically designed products for parenteral administration in pediatric age, carefully following the manufacturer’s recommendations regarding timing and methods of preparation of bags for PN [80].

Parenteral vitamins are usually administered as a mixture of different vitamins. Therefore, the actual dose of vitamins delivered to the patient may be much lower than the intended dose, particularly in the case of retinol (vitamin A) and in premature infants who receive solutions with slow infusion rates.

Currently only a few multivitamin preparations are commercially available for infants and preterm infants. Liposoluble vitamin (A, D, E and K) deficiencies are quite common in preterm infants due to low lipid and fat-soluble vitamins stores and low levels of protein and lipoprotein transport.

Estimates of the vitamin needs of preterm infants are approximate due to lack of adequate studies and the difficulty in adapting the needs of individual cases, which may be affected by many variables, such as gestational age, weight, pre- and postnatal nutritional status (including any maternal deficiencies), clinical conditions, and relationship between parenteral and enteral nutrition (Table 4).

### 3.13. Special Considerations

In this review much attention has been focused on enhancing the nutritional support of very preterm (less than 32 weeks) and extremely preterm (less than 28 weeks) infants. However, it is important to remember that late and moderately preterm (LMPT) infants, that represent the largest population of preterm infants, have specific nutritional requirements, sometimes different from those of the very preterm infant [81]. Further research is needed to determine whether nutritional requirements are mainly dependent on gestational age or birth weight in preterm infants.

A few special considerations are warranted regarding the nutritional management of preterm infants in case of common complications of prematurity, such as PDA, RDS requiring ventilatory support and BPD.

The definition of restrictive or liberal fluid intake varies widely. Evidence suggests that fluid balance in the first days to weeks of life is an important factor influencing neonatal morbidity. Although some authors have recommended restricting the amount of fluid given to preterm infants to allow for a negative water balance in the first few days of life, recently it has been shown that adequate nutrition and caloric intake are crucial in reducing the risk and severity of BPD.

Sung et al. suggested a very restrictive approach with 60 mL/kg on the first day of life to <116 mL/kg/day during the first weeks in preterm infants at high risk of BPD [76]. Other studies defined low intake as <96 mL/kg on the first day of life proceeding to <135 mL/kg/d after the first week. The latter is comparable to the studies included in the Cochrane review by Bell and Acarregui [8], albeit exceeding that by Letshwiti et al. (mild fluid restriction 130–150 mL/kg/d) [77] or the cut-off value of <150 mL/kg/d applied in the Cochrane review by Barrington et al. [78]. Even with regard to the ideal fluid intake for preterm infants with PDA, there is still much debate. PDA is diagnosed in two-thirds of preterm infants with very low birth weight. In the absence of spontaneous duct closure, the conventional management of a hemodynamically significant PDA (hsPDA) is pharmacological treatment, followed by surgical ligation if pharmacological treatment is not successful [79]. In all preterm infants, it is essential to regulate fluid administration to reduce the risk of fluid overload during the first days after birth. Nonetheless, a restriction of fluids, as a first step in PDA management, may subsequently reduce energy and nutrient intake. Infants with RDS feature a greater work of breathing and therefore it is believed that they have a greater energy requirement. Several works in the literature have tried to investigate the best energy source for patients suffering from pulmonary pathology. Glucose intakes greater than 10–12 mg/kg/min should be avoided in patients who have impairment in the elimination of CO_2_. It is important to preserve protein anabolism, as amino acids are responsible for growth and repair of damaged tissues (e.g., alveoli), and for the synthesis of proteins, such as hemoglobin, hormones, and enzymes. As for the recommended lipid mixtures, to date, it is known that long-chain fatty acids have important anti-inflammatory functions and are involved in pulmonary maturation processes [82]. Similarly to preterm neonates with respiratory problems or PDA, the nutritional management of critically ill neonates or surgical infants varies widely: however, in this review we focused our attention only on preterm infants of less than 32 weeks, hence we invite the reader to refer to more specific works on such topics in case of need [83].

## 4. Conclusions

Although early intravenous nutrition for very preterm infants has become standard care, many uncertainties persist about the ideal quantity and balance of single amino-acids, the optimal content of lipid emulsions, the best starting intake and the preferable increase rate of each macronutrient. The new concept of growth for preterm infants, based on growth trajectories of infants instead of fetuses, could have beneficial long-term effects on health. It would therefore be desirable that multicenter randomized controlled trials are designed to explore the effect of early nutrition and growth on long-term health.

## Figures and Tables

**Table 4 nutrients-14-01530-t004:** Recommended doses for parenteral supply of fat soluble and water soluble vitamins for preterm infants [80].

Vitamin A	700–1500 IU/kg/die	Vitamin A plays an essential role in vision, normal differentiation and maintenance of epithelial cells, adequate immune function (T-cell function), reproduction, growth and development.
Vitamin D	200–1000 IU/die	The main function of vitamin D is the regulation of calcium and phosphate. It is essential for bone health. Other health effects of vitamin D, such as prevention of immune-related and infectious diseases, cardiovascular disease, and cancer, have been discussed.
Vitamin E	2.8–3.5 mg/kg/die	Vitamin E (tocopherol) is a lipid-soluble and powerful biological antioxidant which is present in most parenteral lipid emulsions
Vitamin K	10 μg/kg/die	Vitamin K (phylloquinone) regulates carboxylation of the coagulation factors II, VII, IX, X. Protein C and protein S are also vitamin K dependent. Vitamin K plays a role in the synthesis of osteocalcin, a marker of bone formation
Vitamin C	15–25 mg/kg/die	Vitamin C (ascorbic acid) is a cofactor for many enzymes and a strong antioxidant
Thiamine	0.35–0.50 mg/Kg/die	Thiamine pyrophosphate is involved in carbohydrate and lipid metabolism
Riboflavin	0.15–0.2 mg/kg/die	Riboflavin participates in energy metabolism.
Pyridoxine	0.15–0.2 mg/kg/die	Pyridoxine is necessary cofactor for over 100 enzymes that are mostly involved in glycolysis, gluconeogenesis and amino-acid (AA) metabolism, including transamination, deamination, decarboxylation of AA in neurotransmitters (dopamine, serotonin, glutamate, etc.) and the development of the immune system. It is also needed for the synthesis of sphingolipids, hemoglobin and gene expression
Niacin	4–6.8 mg/kg/die	Niacin is essential for the synthesis of nicotinamide adenine dinucleotide and nicotinamide adenine dinucleotide phosphate which serve as cofactors for electron transport and energy metabolism
Vitamin B12	0.3 μg/kg/die	Vitamin B12 is an organometallic complex. It participates in metabolic reactions involving the synthesis of DNA nucleotides
Folic acid	56 mg/kg/die	FA is essential for humans and acts as a cofactor in certain biological reactions; it is needed in the biosynthesis of purines and pyrimidines, for mitotic cell division, in the metabolism of some amino acids and for histidine catabolism
Pantothenic acid	2.5 mg/kg/die	Pantothenic acid (vitamin B5) is required for the synthesis of coenzyme A and therefore essential for fatty acid metabolism.
Biotin	5–8 μg/kg/die

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
