# Peer review of "Macronutrients and Micronutrients in Parenteral Nutrition for Preterm Newborns: A Narrative Review"

_nutrients, 2022, doi:10.3390/nu14071530_

Round 1

Reviewer 1 Report

This review covers and important topic in neonatology, that of appropriate nutrient administration to preterm infants for adequate postnatal growth.

The information that you have in the review is good, but does not seem complete and has some serious deficiencies in structure and organization.

In the first paragraph of the introduction, you copy and paste some lines from your abstract. This could at the least be slightly reworded.

On page two, line 49, you present a hypothesis on the incorrect use of intrauterine growth rates being applied to postnatal preterm infants. In the preceding lines, you make no mention of this, only that it is difficult to achieve adequate growth due to administration of large fluid volumes. These thoughts are not connected.

If you want to include a methods section with a article search, similar to a systematic review, then you need to include the results of that search. How many articles came up in the review, how many were included and excluded etc...

In general, your discussion is very imbalanced. There are very short sections, such as Energy intake, for which there is significant more information that can be given other than a single reference to one paper.

Your paragraphs are broken down into almost bullet points of topics. If your heading says this is a "narrative" review, there should be more discussion.

There is very little reason given for why you only talk about zinc and copper. Either address why these two are of particular focus for you review, or include some information about the other trace elements.

The vitamins section is also very generic and seemingly added very quickly.

Author Response

We thank the reviewer for the high value comments to our work. We have amended the manuscript accordingly. 1. This review covers and important topic in neonatology, that of appropriate nutrient administration to preterm infants for adequate postnatal growth. Thank you very much. 2. The information that you have in the review is good, but does not seem complete and has some serious deficiencies in structure and organization. We revised the entire manuscript according to your opinion in order to make it clearer and more reader-friendly. Moreover, we added some tables to improve the structure of the paper. 3. In the first paragraph of the introduction, you copy and paste some lines from your abstract. This could at the least be slightly reworded. Thank you for your comment. We have reworded the abstract. 4. On page two, line 49, you present a hypothesis on the incorrect use of intrauterine growth rates being applied to postnatal preterm infants. In the preceding lines, you make no mention of this, only that it is difficult to achieve adequate growth due to administration of large fluid volumes. These thoughts are not connected. We agree with the reviewer, hence we deleted the sentence regarding the incorrect use of intrauterine growth charts to assess postnatal preterm infants, since it may sound confounding in this context. 5. If you want to include a methods section with a article search, similar to a systematic review, then you need to include the results of that search. How many articles came up in the review, how many were included and excluded etc... Thank you. However, we believe it is not mandatory to amend the Results. Since the manuscript is a “narrative review”, we opted for a less systematic approach in the Results section. 6. In general, your discussion is very imbalanced. There are very short sections, such as Energy intake, for which there is significant more information that can be given other than a single reference to one paper. Thanks. We have expanded the shorter sections with the aim to rebalance the manuscript. 7.Your paragraphs are broken down into almost bullet points of topics. If your heading says this is a "narrative" review, there should be more discussion. Thank you very much. We have now rewritten the manuscript to make the discussion more evident. 8 There is very little reason given for why you only talk about zinc and copper. Either address why these two are of particular focus for you review, or include some information about the other trace elements. We have added other trace elements to make this section more comprehensive. 9. The vitamins section is also very generic and seemingly added very quickly. Talking about vitamins would imply adding an entire new and complex chapter to this review. Nevertheless, we added a table to sum up major findings about this interesting topic.

Reviewer 2 Report

Parenteral nutrition has been an essential component of care for more than 60 years for preterm infants unable to tolerate volumes of enteral nutrition necessary to meet energy and nutrient requirements.  Yet, the fluid and nutritional needs of preterm remain an active area of research and advances are needed and ongoing.  There have been numerous reviews of clinical trials and previous research, with several in 2021, highlighting the interest in parenteral nutrition for preterm infants.  The following comments can be considered by the authors to improve their contribution. 

  1. There are numerous grammatical errors that detract from the manuscript. The authors should seek assistance from an English speaking individual.
  2. How does this contribution differ from previous reviews of parenteral nutrition for preterm infants?
  3. Lines 50-54: to 64 weeks (gestational age) post-conception is not relevant as this far exceeds the period of interest.
  4. The manuscript would benefit from tables for each macro- and micronutrient that summarize the research that has been done. This will coalesce findings from various studies in a manner that will be easier for readers (and reviewers) to grasp and for the authors to summarize and explain.
  5. The authors describe a search strategy to identify appropriate clinical trials to be reviewed but include information from other reviews and sources.  For example, lines 97-100.
  6. Something that has been largely overlooked in this and previous reviews is how fluid and nutrition support is often adjusted because of other factors. For example, preterm infants with RDS and reliant on mechanical ventilation are commonly provided restricted fluids and diuretics to “protect” the lungs from alveolar flooding.  The authors may be able to make a significant contribution by exploring the responses to PN and fluid volumes with and without mechanical ventilation.  What are the recommendations for infants with respiratory distress syndrome, those requiring mechanical ventilation, or develop bronchopulmonary dysplasia? 
  7. PN solutions and protocols are specific for preterm infants delivered at different gestational ages. For example, low K solutions are recommended for extremely preterm infants until normal micturition.  A description of the gestation specific protocols and PN solutions would be warranted.
  8. The electrolyte information deserves a separate section.
  9. For glucose infusion rates, use the same units (e.g., mg/kg-min or g/kg-day, but not both) to avoid possible confusion.
  10. Lines 187 – 197 include lipid and protein and are not germane to the section about carbohydrates. Best placed in a separate section that could also include the transition from PN to EN. 
  11. Lines 180-182, Does the Stensvold paper describe hypoxia and acidosis with hyperglycermia, or are these secondary to poor gas exchange, thereby limiting glucose metabolism?
  12. Line 200. Reducing to 10.1 g/kg-d is the same as 7 mg/kg-min.  This is where using the same units will avoid confusion. 
  13. The lipid section overlooks the interest and use of lipid emulsions that include fish oils and some of the other advances in this field.
  14. The trace elements section should be expanded to include minerals (e.g., Ca, Mg, and others).
  15. The manuscript would benefit from a section about future research to address gaps in knowledge.

Author Response

We thank the reviewer for the meaningful comments. We have revised the paper accordingly.
1. There are numerous grammatical errors that detract from the manuscript. The authors should seek assistance from an English speaking individual.
We have extensively revised English language.
2. How does this contribution differ from previous reviews of parenteral nutrition for preterm infants?
Thank you for your question. It provides a brief and updated summary on current knowledge regarding parenteral nutrition for preterm neonates. Taking into account that the latest European guidelines (ESPGHAN) trace back to 2018 and few other issues have arisen since then, the present paper aims to fill this gap.
3. Lines 50-54: to 64 weeks (gestational age) post-conception is not relevant as this far exceeds the period of interest.
Thanks. We have amended the sentence accordingly.
4. The manuscript would benefit from tables for each macro- and micronutrient that summarize the research that has been done. This will coalesce findings from various studies in a manner that will be easier for readers (and reviewers) to grasp and for the authors to summarize and explain.
We have added tables to sum up findings for major nutrients.
5. The authors describe a search strategy to identify appropriate clinical trials to be reviewed but include information from other reviews and sources. For example, lines 97-100.
We have amended the methods section by clarifying that we included also observational studies, guidelines and reviews (lines 61-63).
6. Something that has been largely overlooked in this and previous reviews is how fluid and nutrition support is often adjusted because of other factors. For example, preterm infants with RDS and reliant on mechanical ventilation are commonly provided restricted fluids and diuretics to “protect” the lungs from alveolar flooding. The authors may be able to make a significant contribution by exploring the responses to PN and fluid volumes with and without mechanical ventilation. What are the recommendations for infants with respiratory distress syndrome, those requiring mechanical ventilation, or develop bronchopulmonary dysplasia?
We have added a “Special Considerations” paragraph to address these topics.
7. PN solutions and protocols are specific for preterm infants delivered at different gestational ages. For example, low K solutions are recommended for extremely preterm infants until normal micturition. A description of the gestation specific protocols and PN solutions would be warranted.
Thank you for your comment. Indeed, specific requirements for GA are of paramount relevance for preterm infants. However, this aspect would require a further complex paragraph to be properly addressed and this would overcome the aims of this work. Nonetheless, we added a comment regarding this crucial issue in the section “Special Considerations” in order to help the reader find relevant references in case of need.
8. The electrolyte information deserves a separate section.
Thank you very much. We have moved “electrolytes” to a separate section.
9. For glucose infusion rates, use the same units (e.g., mg/kg-min or g/kg-day, but not both) to avoid possible confusion.
We have amended accordingly.
10. Lines 187 – 197 include lipid and protein and are not germane to the section about carbohydrates. Best placed in a separate section that could also include the transition from PN to EN.
Thank you very much for your comment. However, since we mention “aggressive nutrition” in that sentence, which encompasses also high protein and energy intakes by definition, we do not feel like we should make any change in this section.
11. Lines 180-182, Does the Stensvold paper describe hypoxia and acidosis with hyperglycermia, or are these secondary to poor gas exchange, thereby limiting glucose metabolism?
It describes hypoxia and acidosis as a direct consequence of hyperglycemia.
12. Line 200. Reducing to 10.1 g/kg-d is the same as 7 mg/kg-min. This is where using the same units will avoid confusion.
We have amended accordingly.
13. The lipid section overlooks the interest and use of lipid emulsions that include fish oils and some of the other advances in this field.
We have added a new paragraph at the end of the Lipids section to address this interesting topic.
14. The trace elements section should be expanded to include minerals (e.g., Ca, Mg, and others). We have added other trace elements to make this section more comprehensive. 15. The manuscript would benefit from a section about future research to address gaps in knowledge.
Thank you very much. We have added this section in the “Conclusions”.

Round 2

Reviewer 1 Report

I would like to the thank the authors for their responses to my comments and for the additional content they have provided. The content of the article is completely acceptable to me. However, the presentation of the content still requires editing.

There are multiple sections that have paragraphs that are only one - two sentences long. This should be revised to conform with current standards of English grammar. Either these sentences are part of a bullet point list, as you did in the vitamins section, or they need to be expanded or incorporated into a more complete thought in paragraph form. 

There are minor grammatical changes needed:

Page 8 line 395. There is an "o" that needs to be removed.

Author Response

Please see the attachjment

Reviewer 2 Report

I don't consider it to represent a significant improvement or advance of previous reviews, many of which are more extensive and detailed.  A major revision, and with the comments provided, would take longer to complete.  This revision only "touches" on some points and is only a minor improvement. 

Two of the authors are apparently neonatologists and both have published papers about the care of preterm infants.  Yet, they didn't use this knowledge and experience to revise their contribution to provide new and much needed insights into fluid and nutrition support of preterm infants and particularly those delivered at less than 28 weeks.  Notably, initial parenteral nutrition solutions and fluid support for those infants provide very little potassium.  Only after the infants begin to urinate and reduce the excess serum potassium.  This approach reduces the risk of edema.  Also, and as the authors realize, the restriction of fluids for extremely preterm infants is to prevent alveolar "flooding".  

 How can the authors truly advance the field beyond just another review of what is already known?  Incorporate their understanding of how fluid and nutrient strategies of extremely preterm infants differ from those of term infants.  The relationships with organ immaturity. The reality is we know very little about the actual needs of preterm infants, and even the needs of the fetus, and how those are influenced by the transition to the ex utero environment and current standards of care.  
